# Implications of Genomic Newborn Screening for Infant Mortality

**DOI:** 10.3390/ijns9010012

**Published:** 2023-02-28

**Authors:** Monica H. Wojcik, Nina B. Gold

**Affiliations:** 1Divisions of Newborn Medicine and Genetics and Genomics, Department of Pediatrics, Boston Children’s Hospital and Harvard Medical School, Boston, MA 02115, USA; 2Division of Medical Genetics and Metabolism, Department of Pediatrics, Massachusetts General Hospital, Boston, MA 02114, USA

**Keywords:** mortality, infant, neonate, sequencing, exome, genome, genetic, diagnosis, ethics

## Abstract

Technological advances and decreasing costs of genomic sequencing have paved the way for the increased incorporation of genomics into newborn screening (NBS). Genomic sequencing may complement current NBS laboratory analyses or may be used as a first-tier screening tool to identify disorders not detected by current approaches. As a large proportion of infant deaths occur in children with an underlying genetic disorder, earlier diagnosis of these disorders may improve neonatal and infant mortality rates. This lends an additional layer of ethical consideration regarding genomic newborn screening. We review the current understanding of genomic contributions to infant mortality and explore the potential implications of expanded access to genomic screening for infant mortality rates.

## 1. Introduction

Genetic disorders underlie a substantial proportion of infant deaths, particularly infants with congenital anomalies and those admitted to an intensive care unit. Additionally, genetic diagnoses have also been identified in a large proportion of apparently healthy infants who die unexpectedly, although many genetic conditions likely remain undiscovered due to a lack of testing [1,2,3,4]. The early diagnosis of treatable genetic conditions may facilitate access to appropriate therapies. Conversely, the identification of a genetic diagnosis with a poor prognosis may aid families in the decision to withdraw life-sustaining technologies and transition to comfort-focused care [5]. Furthermore, identification of a condition with a high recurrence risk in future pregnancies of the infants’ parents may lead to additional options for reproductive planning, thereby avoiding future infant deaths [4]. Current diagnostic genetic workflows are designed to initiate genetic testing after an infant develops disease symptoms, at which time therapies may not be clinically useful [5,6]. There is increasing interest and an international effort to incorporate genome-wide sequencing into newborn screening approaches, though ethical considerations and other implementation concerns remain unresolved. Here, we comment on the implications of this approach for infant mortality reduction.

## 2. Infant Mortality: The Genomic Landscape

Prior studies have investigated genetic diagnosis in postmortem cohorts [1,3,6,7,8,9,10] and described outcomes that include mortality after diagnostic genetic testing in infants admitted to intensive care units [4,11], as well as mortality outcomes after prenatal diagnosis [12]. These studies have identified varying diagnostic yields that are dependent upon how the cohort was ascertained, with cohorts of sudden, unexpected infant death being identified at a yield of approximately 10% [3,13], while cohorts involving deaths in an intensive care unit setting approach 25–30% [1,4,14].

The spectrum of diagnoses identified also varies by cohort. Genes associated with epilepsy or cardiac arrhythmia are often implicated in cases of sudden, unexpected infant death [3,13] occurring in an apparently healthy infant. Diagnoses identified in cohorts ascertained from intensive care units include multiple malformation syndromes attributed to common aneuploidies or other chromosomal disorders, such as trisomies 13 or 18, or 22q11 deletion syndrome, in addition to monogenic conditions associated with congenital anomalies, severe neurologic conditions, or genetic conditions not typically associated with structural anomalies, such as inborn errors of metabolism [1,15,16].

As these genetic diagnoses are typically identified by a chromosomal microarray or by massively-parallel sequencing technologies, they would be amenable to early detection via genomic sequencing from the dried blood spot obtained for traditional newborn screening, provided appropriate pre-test counseling and consent is obtained [17]. Several genetic conditions, particularly inborn errors of metabolism in addition to spinal muscular atrophy, are already included in newborn screening panels in many programs in the United States that have a selective approach, where conditions are identified for inclusion in newborn screening panels based upon particular criteria—ideally, conditions for which early treatment is available and leads to meaningful improvements [18]. However, many additional conditions leading to death that were once not treatable may now be amenable to precision treatments or other targeted therapy approaches, particularly as anti-sense oligonucleotide and gene therapies are rapidly emerging [19].

## 3. Current Barriers to Understanding

Our understanding of the depth and breadth of genetic conditions responsible for infant deaths is limited by several factors. First and foremost, the lack of population-based approaches for a comprehensive genomic evaluation of infant deaths limits our abilities to quantify the public health impact of these diagnoses. Conclusions identified from current cohorts are therefore limited by selection bias. In addition, the interpretation of variants identified is limited in the perimortem setting due to a lack of ability for follow-up investigations. Thus, many infants with genetic conditions are never identified, and the experience with genetic diagnosis in pediatric populations suggests that this may disproportionately impact historically underserved populations, though further research is needed into inequities in this realm [20,21]. Thus, this is a public health concern with particular bioethical overtones.

Finally, limited outcome data preclude accurate estimates of mortality rates for conditions that are identified in the perinatal setting. Attempts to quantify these mortality rates have been undertaken for specific diseases, such as genetic leukodystrophy syndromes [22], though accurate estimates are limited by challenges in death reporting, where specific genetic conditions are difficult to identify [23].

## 4. Impact of Genomic Newborn Screening on Infant Mortality

Genomic newborn screening has the potential to reduce infant mortality by identifying infants with treatable diseases prior to the onset of irreversible symptom progression, leading toward the improved management of neonates and infants with a range of genetic disorders, although the spectrum of possible impacts is not currently well understood. If applied on a population-wide scale, genomic newborn screening techniques may allow for more a comprehensive description of genetic diagnoses associated with infant deaths by eliminating the inherent bias in the access to a clinical diagnostic genetic evaluation, where many infants die before genetic diagnoses can be identified, as our current knowledge of the prognosis for these conditions is biased toward those who survive long enough to have genetic testing. Although prior research suggests that genomic sequencing is a robust method for the detection of treatable conditions [24], issues related to ethical implementation, particularly informed consent, remain to be fully addressed. If newborn screening is not fitting as a system to incorporate this type of genetic screening due to constraints related to costs and timely results, broad genomic sequencing may be introduced as routine outside of newborn screening: e.g., tailored to the newborn population with emerging health issues.

Nonetheless, dedicated efforts must be undertaken to move toward equitable implementation if genomic newborn screening is to be introduced. Additionally, disorders may be identified for which there are current or emerging therapies that lead to improved survival, though, again, ensuring equitable access to such therapies is paramount to upholding the ethical principle of justice in healthcare. Finally, conditions can be identified that have implications for parents’ future childbearing, and the identification of a precise genetic diagnosis can aid in pregnancy planning that can further reduce neonatal and infant mortality.

## 5. Conclusions

Genomic newborn screening presents a unique opportunity not only to identify and manage infants at risk for long-term medical sequelae of a wider range of underlying genetic conditions, but also to understand and potentially reduce rates of infant mortality. Resource availability regarding the broader application of genomic newborn screening remains a valid concern and area of further focus. Still, understanding this approach and its potential as an important public health outcome of infant mortality may provide additional ethical justification. Critical to the success of this approach is the equitable implementation of genomic newborn screening in addition to resources devoted to accurately capturing health outcomes.

## Data Availability

Not applicable.

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
