# Peer review of "Implications of Genomic Newborn Screening for Infant Mortality"

_2409-515X, 2023, doi:10.3390/ijns9010012_

Round 1

Reviewer 1 Report

Sentence beginning on line 59 - "...in a more targeted approach"... what are the author's trying to convey? There are 62 conditions recommended for screening in the United States - are they saying that with genomic sequencing we can detect these 62 plus many others. Suggest changing "targeted approach" and add in the information that conditions in the US generally have an evidence base that indicates screening will lead to decrease in mortality and morbidity and that there is a process to decide what to screen for. Many of the conditions identified by genomic sequencing don't have treatments.

Author Response

Reviewer 1: Sentence beginning on line 59 - "...in a more targeted approach"... what are the author's trying to convey? There are 62 conditions recommended for screening in the United States - are they saying that with genomic sequencing we can detect these 62 plus many others. Suggest changing "targeted approach" and add in the information that conditions in the US generally have an evidence base that indicates screening will lead to decrease in mortality and morbidity and that there is a process to decide what to screen for. Many of the conditions identified by genomic sequencing don't have treatments.

Author response: Thank you for suggesting this clarification. We have now revised the paragraph beginning on line 59, which now reads: “Several genetic conditions, particularly inborn errors of metabolism in addition to spinal muscular atrophy, are already included on newborn screening panels in many programs in the United States in a selective approach, where conditions are identified for inclusion in newborn screening panels based upon particular criteria – ideally, conditions for which early treatment is available and leads to meaningful improvements. However, many additional conditions leading to death that were once not treatable may now be amenable to precision treatments or other targeted approaches at therapy, particularly as anti-sense oligonucleotide and gene therapies are rapidly emerging [17]."

Reviewer 2 Report

The paper by Wojcik and Gold provide a commentary on the potential for genomic sequencing in the newborn period to impact infant mortality rates.

While the paper provides a reasonable argument and evidence for how newborn genomic sequencing may reduce infant mortality rates, the authors miss an opportunity to discuss the important differences between public health newborn screening programs and clinical genomic sequencing in the newborn population. These two distinct programs continue to be conflated in the literature and in discussions around integration of genomics in healthy newborns. Likewise, the real ethical issues of mandating genomic sequencing versus consenting in a clinical setting and the decisions of what variants are or are not reported as they pertain to issues of equity are glossed over.

More targeted feedback is below:

Line 25-27: Clarify sentence. It begins by saying that genetic diagnoses have been identified in a large proportion of unexpected deaths, but then says that they likely remain undiagnosed. I believe this is meant to say that many cases of unexpected deaths who did not have diagnoses likely had genetic causes as well?

Lines 47-55: These sentences are very long and are difficult to follow. Please reword for clarity.

Line 56-59: Given ongoing and recent issues around the use of DBS specimens after NBS and concerns regarding privacy, I'm concerned with the assertion that this could just simply be done from the same DBS obtained for traditional NBS. Especially since consent is not obtained for traditional NBS. It is fair to say that the same matrix (DBS) that is used for traditional NBS could be used for newborn genomic sequencing as well, with a citation illustrating that WGS can be done in DBS.

Lines 72-75: I certainly agree that there are disparities in access to genetics and in genetic diagnoses in underserved populations; however, this makes a leap from what we see clinically to what is happening peri-mortem. Do we know whether this disparity is retained when examining rates of genetic diagnoses peri-mortem? It is likely that it does, but I think this needs to be written in a way that this is being extrapolated from what is seen clinically in pediatrics.

Line 85-88: This is confusing as, in theory, if genomic sequencing is being done for diseases that are treatable, then we should see a decrease in infant deaths from these diseases, yes? Or is this saying that we may have better birth prevalence estimates of diseases that prior to sequencing lead to infant deaths? And again, is the bias in access to a clinical diagnostic genetic evaluation the same as bias in peri-mortem access to genomic sequencing when these infants are likely in an ICU?

Author Response

Reviewer 2: The paper by Wojcik and Gold provide a commentary on the potential for genomic sequencing in the newborn period to impact infant mortality rates. While the paper provides a reasonable argument and evidence for how newborn genomic sequencing may reduce infant mortality rates, the authors miss an opportunity to discuss the important differences between public health newborn screening programs and clinical genomic sequencing in the newborn population. These two distinct programs continue to be conflated in the literature and in discussions around integration of genomics in healthy newborns. Likewise, the real ethical issues of mandating genomic sequencing versus consenting in a clinical setting and the decisions of what variants are or are not reported as they pertain to issues of equity are glossed over.

Author response: Thank you for these comments. We wholeheartedly agree with the issues that the author raises about the distinction between the current newborn screening programs and potential future prospects integrating genomics. While that nuanced discussion is outside of the scope of this commentary, we have revised the manuscript taking into consideration the issues that the reviewer mentions.

More targeted feedback is below:

Line 25-27: Clarify sentence. It begins by saying that genetic diagnoses have been identified in a large proportion of unexpected deaths, but then says that they likely remain undiagnosed. I believe this is meant to say that many cases of unexpected deaths who did not have diagnoses likely had genetic causes as well?

Author response: Thank you for suggesting this clarification. We have revised this sentence, which now reads as follows: “Additionally, genetic diagnoses have also been identified in a large proportion of apparently healthy infants who die unexpectedly, although many genetic conditions likely remain undiscovered due to a lack of testing.”

Lines 47-55: These sentences are very long and are difficult to follow. Please reword for clarity.

Author response: Thank you for suggesting this revision. We have reworded the paragraph, which now reads: “The spectrum of diagnoses identified also varies by cohort. Genes associated with epilepsy or cardiac arrhythmia are often implicated in cases of sudden, unexpected infant death [3, 13], occurring in an apparently healthy infant. Diagnoses identified in cohorts ascertained from intensive care units include multiple malformation syndromes attributed to common aneuploidies or other chromosomal disorders, such as trisomy 13 or 18 or 22q11 deletion syndrome, in addition to monogenic conditions associated with congenital anomalies, severe neurologic conditions, or genetic conditions not typically associated with structural anomalies, such as inborn errors of metabolism.”

Line 56-59: Given ongoing and recent issues around the use of DBS specimens after NBS and concerns regarding privacy, I'm concerned with the assertion that this could just simply be done from the same DBS obtained for traditional NBS. Especially since consent is not obtained for traditional NBS. It is fair to say that the same matrix (DBS) that is used for traditional NBS could be used for newborn genomic sequencing as well, with a citation illustrating that WGS can be done in DBS.

Author response: Thank you for this clarification. We have revised this statement as follows and added a citation as suggested: “As these genetic diagnoses are typically identified by chromosomal microarray or by massively-parallel sequencing technologies, they would be amenable to early detection via genomic sequencing from the dried blood spot obtained for traditional newborn screening, provided appropriate pre-test counseling and consent is obtained.”

Lines 72-75: I certainly agree that there are disparities in access to genetics and in genetic diagnoses in underserved populations; however, this makes a leap from what we see clinically to what is happening peri-mortem. Do we know whether this disparity is retained when examining rates of genetic diagnoses peri-mortem? It is likely that it does, but I think this needs to be written in a way that this is being extrapolated from what is seen clinically in pediatrics.

Author response: Thank you for suggesting this clarification. We have revised this statement, which now reads: “Thus, many infants with genetic conditions are never identified, and experience with genetic diagnosis in pediatric populations suggest that this may disproportionate impact historically underserved populations, though further research is needed into inequities in this realm[20, 21].”

Line 85-88: This is confusing as, in theory, if genomic sequencing is being done for diseases that are treatable, then we should see a decrease in infant deaths from these diseases, yes? Or is this saying that we may have better birth prevalence estimates of diseases that prior to sequencing lead to infant deaths? And again, is the bias in access to a clinical diagnostic genetic evaluation the same as bias in peri-mortem access to genomic sequencing when these infants are likely in an ICU?

Author response: Thank you for these comments. We have clarified this section to reflect that genomic sequencing may reduce infant deaths in addition to clarifying the mortality burden of genetic conditions, as well as to clarify that the nature of this bias is not fully understood as the population burden of genetic disorders in this age group is not known. This section now reads as follows: “Genomic newborn screening has the potential to reduce infant mortality by identifying infants with treatable diseases prior to the onset of irreversible symptom progression, leading toward improved management of neonates and infants with a range of genetic disorders, although the spectrum of possible impact is not currently well understood. If applied on a population-wide scale, genomic newborn screening techniques may allow for more a comprehensive description of genetic diagnoses associated with infant deaths by eliminating the inherent bias in access to a clinical diagnostic genetic evaluation, where many infants die before genetic diagnoses can be identified and thus our current knowledge of the prognosis for these conditions is biased towards those who survive long enough to have genetic testing.”

Reviewer 3 Report

With the advances in next-generation sequencing technology and computational sciences, genomic newborn screening technology is now applying to identify genetic causes underlying human disorders and thereby has potential to reduce rates of infant mortality. Overall, this manuscript is in time and well-written. It is qualified to publish in IJNS.

I have two comments.

1. As claimed by the authors in line 82, NBS has potential to reduce rate of infant mortality for treatable diseases prior to the onset of irreversible symptom progression. It is not clear to the reader which are those treatable diseases. How many or what is the proportion of such treatable diseases in the context of new born disorders? It will provide clues how large the impact of genomic newborn screening on infant mortality. Moreover, the authors are suggested to provide some examples or give some references for such treatable diseases. 

2. Since most of the genetic disorders contributed to infant mortality maybe rare diseases, it is challenging in NBS to identify rare variants underling disorders due to the sequencing errors or other factors. The authors are suggested to mention such limitation in NBS somewhere in the manuscript? 

line 25, get rid of the extra comma.

Author Response

Reviewer 3:With the advances in next-generation sequencing technology and computational sciences, genomic newborn screening technology is now applying to identify genetic causes underlying human disorders and thereby has potential to reduce rates of infant mortality. Overall, this manuscript is in time and well-written. It is qualified to publish in IJNS.

I have two comments.

  1. As claimed by the authors in line 82, NBS has potential to reduce rate of infant mortality for treatable diseases prior to the onset of irreversible symptom progression. It is not clear to the reader which are those treatable diseases. How many or what is the proportion of such treatable diseases in the context of new born disorders? It will provide clues how large the impact of genomic newborn screening on infant mortality. Moreover, the authors are suggested to provide some examples or give some references for such treatable diseases. 

Author response: Thank you for this important and valid question. Unfortunately, the proportion is unknown, and we have clarified this statement as follows: “Genomic newborn screening has the potential to reduce infant mortality by identifying infants with treatable diseases prior to the onset of irreversible symptom progression, leading toward improved management of neonates and infants with a range of genetic disorders, although the spectrum of possible impact is not currently well understood.”

  1. Since most of the genetic disorders contributed to infant mortality may be rare diseases, it is challenging in NBS to identify rare variants underling disorders due to the sequencing errors or other factors. The authors are suggested to mention such limitation in NBS somewhere in the manuscript? 

Author response: These are rare conditions, though our prior work has suggested that sequencing may be a robust approach. We have revised our discussion of the impact of genomic newborn screening to include the following: “Although prior research suggests that genomic sequencing is a robust method for detection of treatable conditions [24], issues related to ethical implementation, particularly informed consent, remain to be fully addressed.”

line 25, get rid of the extra comma.

Author response: Thank you – we have removed this.